# Real Time Water-In-Oil Emulsion Size Measurement in Optofluidic Channels

**DOI:** 10.3390/s22134999

**Published:** 2022-07-02

**Authors:** Juliana N. Schianti, Igor Y. Abe, Marco I. Alayo, Daniel O. Carvalho

**Affiliations:** 1Electrical Engineering Department (ENE), University of Brasilia, Brasilia 70910-900, Brazil; juliana.schianti@alumni.usp.br; 2Polytechnic School, University of São Paulo (USP), São Paulo 05508-010, Brazil; igor.abe@usp.br (I.Y.A.); malayo@usp.br (M.I.A.)

**Keywords:** emulsion, size measurement, optofluidics

## Abstract

In this work, we investigated a platform for real-time emulsion droplet detection and size measurement in optofluidic platforms. An 8.2 µm core diameter input optical fiber and a multi-mode Gradient Refractive Index (GRIN) output fiber were integrated into an acrylic microfluidic channel platform consisting of three layers. Water-in-oil emulsions were investigated, since relevant applications have emerged in the recent past for these types of emulsions, such as drug encapsulation as well as droplet-based Polymerase Chain Reaction (PCR) amplification of DNA, among others. The main contribution of this work is in understanding the main physical phenomena (i.e., total internal reflection, refraction, and interference) behind the complex transmittance pattern obtained for these droplets. For this purpose, a frequency domain electromagnetic wave propagation modelling of the structure using the Finite Element Method (FEM) was used along with experimental measurements.

## 1. Introduction

Naturally occurring emulsions, such as milk and butter, have been consumed and produced by the human species for a very long time. On the other hand, only in the past several decades, important scientific and pharmaceutical applications of these types of dispersion have emerged. Emulsions are composed of a liquid phase dispersion of single or multiple droplets suspended in an immiscible continuous phase. In this sense, a very relevant application of emulsions is for encapsulating drugs in shell-like structures used in smart drugs. With the appropriate choice of polymeric materials, these drugs can be made to deliver specific chemical agents, used for therapeutic purposes, directly to an intended site of action inside the human body, where the protecting shell would dissolve due to the combined effects of pH and temperature, for example. These kinds of structures could also be used for controlled drug release depending on the encapsulating material and dissolution rate [1,2].

More recently, microfluidic devices have been shown to enable emulsion production with much more precise control over droplet size. Monodisperse emulsions can be produced using microchannels, meaning that sets of droplets with much greater uniformity in terms of size are obtained. Fluidic platforms have been demonstrated for encapsulating cells [3], viruses and bacteria [4], nanoparticles [5], as well as flavours and vitamins [6]. Besides allowing great control over size and uniformity, microfluidics droplet production is also benefited by other key aspects of this technology, such as small reagent consumption associated with the small fluid flow rates that are typically used, small energy consumption, and high level of control over physical and chemical properties of the fluids used.

A key enabling technology for which microfluidic droplets have been applied with success is the Polymerase Chain Reaction (PCR), pervasively used in DNA analysis nowadays [7,8]. Recently, microfluidic platforms in which the entire PCR process is realized entirely on chip have been proposed [9,10,11,12,13,14]. Usually, the aqueous droplets are formed inside a polar (oil) continuous phase, which prevents contamination with spurious DNA strands since the solution does not come into contact with the internal walls of the reservoir or microchannel. The detection limit for DNA analysis is much smaller than in bulk solutions and the size of the water-in-oil droplets used directly affects this detection limit.

Be it for PCR and DNA analysis, or for other applications such as drug delivery, the ability to have means for real-time measurement and even control over droplet size is a very desirable feature. Different techniques have been proposed for real-time droplet detection and size measurement, such as: methods that rely on the change of capacitance [15] or impedance [16] of the liquids flowing through the microchannels; radiofrequency (RF) based methods, in which a variation of the permittivity of the liquids induces a change in the phase [17] or amplitude (typically using resonators) [18] of electromagnetic wave flowing through the liquids; image analysis methods, in which the images captured with a high-speed camera are analyzed in real-time [19]; optical methods using light reflected or transmitted through the sample [20].

In this work, we investigated an optical platform for droplet detection and size measurement in real-time, which could also potentially be used for droplet size control if the measured dimensions were used in a feedback mechanism to control the flow of the continuous and dispersed phase fluids. A laser and a photodiode were used to generate and detect the optical power transmitted through the droplets as they flow along the microchannels. We used an 8.2 µm core diameter input optical fiber in order to couple the signal to the device and a Graded Index (GRIN) multimode optical fiber to capture the light transmitted through the droplets. Because the core diameter of both the input fiber (8.2 µm in diameter) and output fiber (50 µm in diameter) were significantly smaller than the ones used in previous works (~100 µm in diameter) [20], we presumed that the investigated device could be used to measure smaller droplets, although this cannot be backed by experimental data since the droplet diameter is not specified in that work. The main novelty and contribution of this work are in understanding the main mechanisms and physical phenomena behind the transmittance pattern obtained for these droplets. For this purpose, frequency domain Electromagnetic wave propagation modelling of the structure using the Finite Element Method (FEM) was used along with experimental measurements.

## 2. Device Fabrication

The layers of the device used to measure droplet dimension are illustrated in Figure 1a. The top and bottom layers are 3 mm-thick acrylic sheets, which are used to seal the microchannel. The only pattern in these layers consists of the three holes through the top layer which serves the purpose of letting the water (dispersed) and oil (continuous) phase into the microchannel and the emulsion out of this device. Acrylic was chosen for being a hydrophobic material, which allows the generation of water-in-oil droplets. If a hydrophilic material, such as glass were used, oil-in-water emulsions could be generated in a similar fashion but considering the aimed future applications mentioned in the introduction (PCR, DNA analysis), we opted for water droplets in a continuous phase.

The middle channel layer is a 300 µm-thick sheet of acrylic with the microchannel geometry (T-junction) patterned on it. Grooves for the input and output optical fibers are also patterned on this layer. All geometries were patterned into the acrylic sheets with a CO_2_ laser. The width of the longer main channel (continuous phase) is 500 µm and the width of the shorter (dispersed phase) channel in the T-junction is 250 µm. The roughness associated with the channel sidewalls was estimated to be around 13 µm using image analysis. This is not a big problem for our system since both input and output fibers are positioned in such a way that light impinges normally upon these surfaces, and the fraction of power which is actually scattered does not seem to be large, since the power at the output fiber is of the order of hundreds of µWatts and the laser diode power is 1 mW. For different ratios of continuous and dispersed phase flow rates, the dispersed phase is cut off by the continuous phase, forming droplets with diameters that depend on the ratio of the flow rates associated with both phases. By varying the ratio of these flow rates, it is therefore possible to obtain droplets with different dimensions. The three layers shown in Figure 1 were bonded using UV curing adhesive (Loxeal, 30-21), which was exposed to light from a 36 W UV lamp for several minutes.

Three brass connectors (Figure 1a) are used in order to couple plastic tubes which bring the dispersed and continuous phase fluids from two syringe pumps into the microchannel, as well as the generated emulsion into and outside the reservoir.

The input optical fiber has a core diameter of 8.2 µm and the output optical fiber is a GRIN fiber with a 50 µm-thick core. The input fiber is connected to the laser using a PC connector and it would be possible to splice fibers with different types of connectors, such as PC connectors, to both these fibers in order to use lasers and detectors of different types.

## 3. Experimental Setup and Results

In order to produce the water-in-oil emulsion, the T-junction devices discussed in the previous section were used. Two syringe pumps (Injectomat MC Agilia, Fresenius Kabi, Paris, France) were used to control the continuous phase (soybean oil) and dispersed phase (de-ionized water) fluid flows. As shown in Figure 1b the oil phase was introduced in the main channel and the water phase was introduced in the smaller lateral channel of the T-junction. In this way, the water phase is cut off by the continuous phase in a periodic fashion in such a way that the droplets are formed. The oil phase flow was varied from 5.68 mL/h up to 9.23 mL/h in order to obtain droplets of different sizes. In all experiments, the water (dispersed phase) flow was maintained at 0.1 mL/h. A laser diode emitting 1 mW of optical power at 632.8 nm, coupled to an optical fiber with a core diameter of 8.2 µm (SMF-28) was used to couple light to one side of the micro-channel (input fiber in Figure 1a). The light transmitted through the channel (and the droplets flowing through it) was captured by a GRIN optical fiber with a core diameter of 50 µm (output fiber in Figure 1a). The light from this fiber was detected using a photodiode coupled to a trans-conductance amplifier (OPT101 chip from Texas Instruments) and detected using LabVIEW. Figure 2 shows the transmitted power curves for the six different flow values associated with the continuous (oil phase) flow as a function of time. The reason for the power fluctuation observed when the droplet is not in the optical path (baseline fluctuation) has to do with the large dependence of the power emitted by the laser diode with respect to temperature. This is why, for the six different measurements shown in Figure 2, the horizontal line fluctuates between 700 and 900 a.u. between one measurement and the other. This could be addressed by temperature control of the emitted power. 

It can be readily seen in these figures that when the droplets flow through the optical path, starting at the input fiber and ending at the output fiber, there is a drop in transmitted power (upside-down M-shaped patterns). Furthermore, with the decrease of the continuous phase flow, the width (time-duration) of these M-shaped patterns becomes larger, indicating an increase in droplet diameter. The overall shape of the transmitted power curves when the droplets cross the optical path might be unexpected at a first glance, but a few optical phenomena play a role in the different parts of these upside-down M-shaped curves. In order to understand this behavior, we will rely on the electromagnetic modelling discussed in the following section. One aspect which is noteworthy is that, once the edge of the droplet gets in the way of the light beam there is a very abrupt drop in the transmitted power, the same being true for the situation when the droplets exit the path of the optical beam. As we will see over the next section, this is associated with the fact that, at these positions, the main optical phenomenon that takes place is Total Reflection which deflects the beam to a position far away from the output fiber.

The reason why the graphs in Figure 2 present different amplitudes of the M-shaped patterns, even for a normalized transmitted power, is attributed to the fact that, especially for smaller droplets, it is impossible to control the droplet position along the out-of-plane (z-direction) with respect to the microchannel surface, in our device. For this reason, for some of the samples, the beam might go through a point that is much closer to the ‘equator’ of the droplet, and for others, the transmitted beam might go through a point that is farther from the equator. In the latter case, due to refraction, the beam would be deflected vertically, away from the output fiber, creating a drop in the transmitted power. To account for this in electromagnetic modelling would require full 3D-simulations, which unfortunately we are not able to perform due to limitations in our computational resources. For this reason, as will be discussed in the next section, we have performed 2D simulations in order to understand these phenomena. It would also be possible to have this sensor work in the reflected power regime, since we used normal incidence, but this would require additional components in the setup, such as a circulator or at least a directional coupler connected at the input power to separate the reflected mode from the reflected mode.

Figure 3 shows the droplet diameter measured using optical microscope images of droplets collected at the output of the micro-channel as a function of the Full Width at Half-Maximum (FWHM) of the upside-down M-shaped curves shown in Figure 2. Points (1) through (6) in the graph correspond to curves with increasing continuous phase flows in Figure 2 (starting at the black curve and ending at the red one). The insets in this figure show optical microscope images of droplets with different sizes corresponding to each point in the graph.

The FWHM shown in the abscissa of the graph is de-normalized with respect to the flow velocity associated with each continuous phase flow adopted. This is done by dividing the FWHM of the M-shaped curves shown in Figure 2 by the flow velocity associated with each continuous phase flow (from 9.23 mL/h down to 5.68 mL/h). This is necessary since, with the increase of flow values, the increasing speeds with which the droplets cross the light path would decrease the FWHM, not because the droplet itself is smaller but rather because the time of interaction with the beam is smaller. As it can be seen, there is a linear relationship between the de-normalized FWHM with respect to droplet diameter, which can be used in order to detect the diameter in real-time and even implement some sort of control mechanism, by varying the flow values in order to produce droplets with a desired diameter.

## 4. Electromagnetic Modelling and Discussion

Frequency domain electromagnetic wave propagation modelling of the droplets moving across the path of the beam going from the input fiber to the output fibers was performed using the Finite Element Method (FEM) using the commercial software Comsol Multiphysics. This allowed us to obtain a much better grasp of the physical phenomena taking place. Figure 4a shows the results of fifteen simulations in which the position of the droplets with respect to the input/output fiber was varied in steps of 20 µm. For this particular set of simulations, the diameter of the droplet was 325 µm, but different diameters were modelled in order to better understand the experimental results.

In all the simulations, Perfectly Matched Layers (PMLs) were used at the borders of the simulation domain. The mesh was refined until the results no longer varied and the wavelength of light matched the one used experimentally λ = 632.8 nm. A field distribution associated with the fundamental mode of the 8.2 µm core diameter input fiber was set up as excitation at the bottom of each simulation domain (see field distributions in Figure 4a). Simulations were also performed with higher modes for the input optical fiber, but the results were similar to those obtained for the fundamental mode. The average Poynting vector in the direction of propagation was integrated over a window with lateral dimensions of 50 µm (matching the core of the output fiber) at the opposite side of the computational domain (where the interface with the output fiber would be in the experiments).

In order to present the results in a more concise way, Figure 4a shows, for each position, only a narrow strip containing the fields in the modelled domain. This corresponds to the part in which the physical phenomena take place. Figure 4b is a 3D illustration showing the position of the input/output fibers, which are always aligned, with respect to the entire droplet, for three different positions shown in Figure 4a: position (1), in which the droplet is immediately to the right of the beam going from the input to the output fiber; position (8) in which the beam goes straight through the center of the spherical droplet; position (15) in which the droplet is immediately to the left of the light beam. The interaction of the light beam with the droplet in all intermediary positions (in between 1 and 8, as well as in between 8 and 15) is shown in Figure 4a, which shows the magnitude of the electric field associated with the beam of light.

Different optical phenomena are observed in Figure 4a. As the beam propagates from the input fiber on the bottom, to the output fiber on top, it goes through diffraction and its initial waist diameter (around 10 µm) ends up at around 50 µm at the position of the output fiber. As the droplet starts (and stops) to get in the way of the beam, the angle of incidence of the beam with respect to normal to the interface between the oil phase (outside the droplet) and the water phase (droplet) is very large and the beam of light goes through Total Reflection, being totally deflected to a direction far away from the output fiber. Total Reflection (similar to Total Internal Reflection inside an optical fiber’s core) is possible here because the incidence medium (oil) has a higher refractive index (*n_oil_* = 1.46) than that of the droplet medium (*n_water_* = 1.33). This is clearly seen in positions (2) and (14) of Figure 4a.

At position (3), the beam suffers a double refraction: the first going into the droplet medium (water) and another coming out of the droplet. Note that, due to the refractive indices of both media, the refraction at the first interface deflects the beam away from the detector, and the refraction at the second interface has the same effect. As the droplet moves towards the left, from position (3) to position (7), the beam is deflected less and less by both these refractions and moves closer and closer to the output fiber positioned at the center top of each figure. Note that the beam deflections associated with positions (9) through (14) are symmetrical with respect to the beam reflections associated with positions (7) through (3).

As the beam goes through the center of the droplet at position (8), it is not steered away from the output fiber by the refractions at both interfaces. On the other hand, another phenomenon takes place here, which can be observed only by looking at the interaction of light with droplets of different diameters. The relevant phenomenon, in this case, is interference, since, at normal incidence, the interfaces at either side of the droplet are partially reflective. Therefore, the droplet, right at this position, behaves as an optical cavity, with a very poor-quality factor, because of the low reflectivities (Γ ≈ 0.002). As a result of this, one would expect that, depending on the diameter of the droplet, which would correspond to the length of the cavity, there should be a minimum or a maximum in transmission.

The transmittance of the device was calculated by dividing the power captured by the output fiber by the input power. The output power, as stated, was obtained from the average Poynting vector’s component in the direction of propagation integrated over a window with lateral dimensions of 50 µm at the upper side of the computational domain (refer to Figure 4a). By varying the position of droplets with different sizes, the graphs shown in Figure 5 were obtained. The six different curves correspond to droplets with diameters ranging from 50 µm (black) up to 400 µm (red). By observing the curves corresponding to the largest diameters (red and orange), it is noteworthy that as the droplet gets in the way of the beam, there is a very abrupt drop in transmittance, which can be seen on either side of these curves, due to symmetry. These drops in transmittance correspond to positions (2) and (14) in Figure 4a, in which the beam goes through Total Reflection at the droplet edges, and the beam is suddenly deflected to a position very far from the output fiber.

As the droplet moves further into the beam’s path, the mechanism responsible for deflecting the beam transitions from Total Reflection into a double refraction (at different positions on the surface of the droplet). Because of this transition, there is a small oscillation in the transmittance curve at positions that are close to the edge of the spectra moving inwards towards the center. Beyond this oscillation, the double refraction mechanism steers the beam closer and closer to the output fiber. This is why, around the center of the transmittance curve (droplet position = 0.0 µm), the transmittance rises (or falls), in a less abrupt fashion, and there is an inflection point right at the central position. Here it should be clear that the portion of the curves, around the central region, where the transmittance rises and falls almost linearly correspond to positions (3) through (7) and (9) in Figure 4a.

The inflection point at the center of each transmittance curve corresponds to position (8) in Figure 4a. In some of the curves, local maxima can be observed at this position. When the beam is centered with respect to the droplet, the interfaces do not deflect the beam, which explains these local maxima. On the other hand, for some curves (red, green, and blue curves) the central point does not correspond to a local maximum. On these curves, the central position corresponds to local minima. This is attributed to the already mentioned fact that, right at this position interfaces at either side of a given droplet constitute, due to being partially reflective, make up a low Q-factor cavity. Therefore, depending on the length of the cavity (diameter of the droplet) the transmitted light will suffer either constructive or destructive interference inside the cavity. These conditions would correspond to a local maximum (orange, purple, and black curves) or a local minimum (red, green, and blue curves) in transmittance, respectively. The minimum droplet that was detectable in our simulations was 10 µm, which corresponds to dimensions close to the beam diameter exiting the input fiber.

## 5. Conclusions

We have presented a device for real-time optical measurement of droplet dimensions in microfluidic channel-based emulsion generation platforms. Droplets with dimensions ranging from 60 µm to 325 µm have been generated by varying the flow ratios in a conventional T-junction microchannel and measured optically using the transmittance signal. The most important contribution of this work is a deeper understanding of the mechanisms and physical phenomena underlying the transmitted power curve obtained when the droplets cross the optical pathway. As we have shown, refraction, interference (optical cavity effect) and Total Reflection play an important role in the shape of these curves. The latter (Total Reflection) is responsible for the abrupt drop in transmittance at the edge of the upside-down M transmittance patterns. The abrupt transition at the edges of such curves makes it possible for the droplet diameters to be measured more accurately, when compared to other methods in which the transition from a high transmittance state to a low transmittance state is less abrupt (smaller slope in these regions of the curve). Future works could deal with the problem of implementing real-time control over the generated droplet dimensions, as well as using the relatively complex shape of the different curves shown in Figure 5 to implement an even more accurate characterization of the droplet characteristics, which could perhaps be done with the aid of artificial intelligence algorithms.

## Figures and Tables

**Figure 1 sensors-22-04999-f001:**
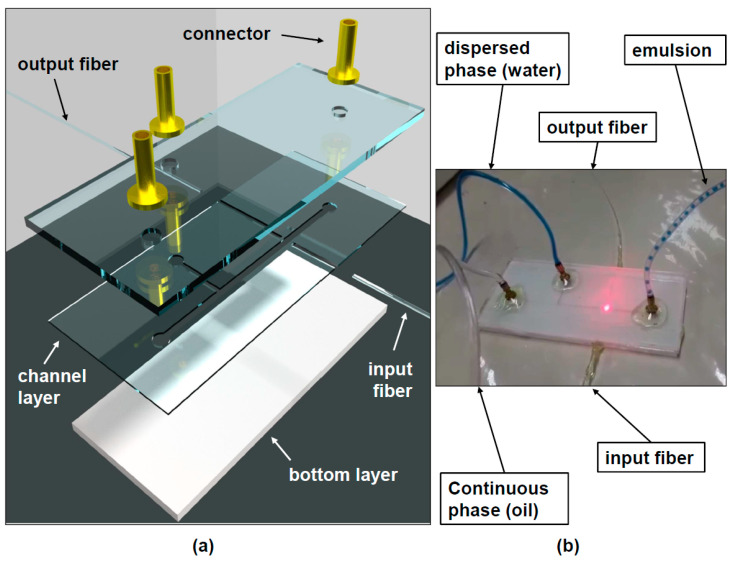
(**a**) Illustration of the fabricated device and (**b**) actual device generating emulsion (blue dye was added to the water phase, just for this picture, in order for the droplets to be visible).

**Figure 2 sensors-22-04999-f002:**
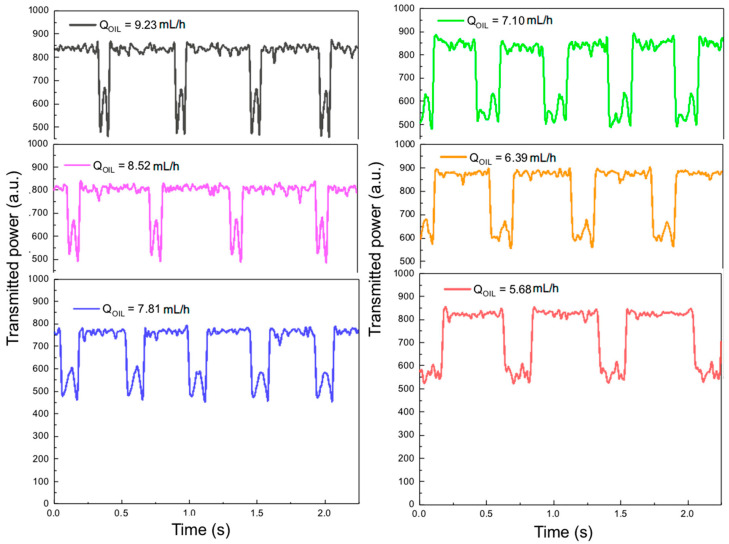
Transmittance curves obtained experimentally as a function of time for different flow rates.

**Figure 3 sensors-22-04999-f003:**
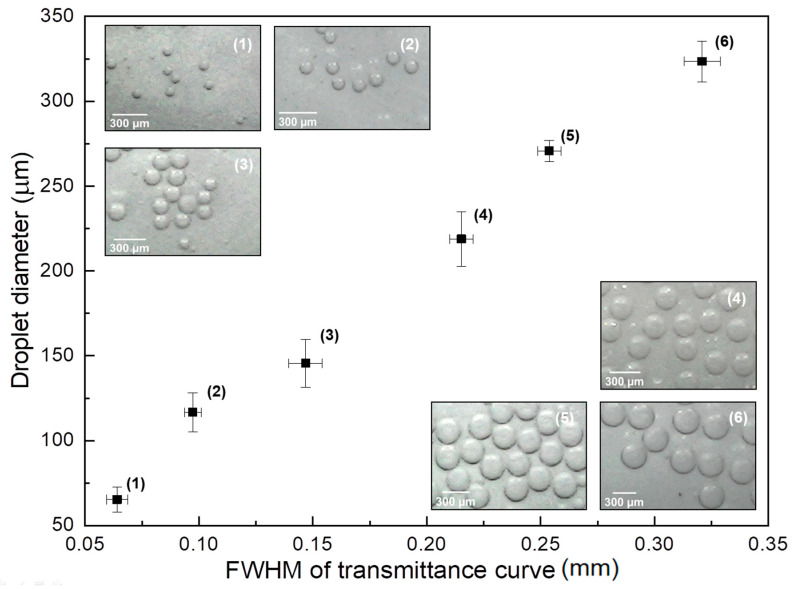
Droplet diameter, measured with optical microscope, as a function of the Full Width at Half-Maximum of the transmittance curves (Figure 2). The numbered insets show optical microscope images of droplets corresponding to each point in the graph.

**Figure 4 sensors-22-04999-f004:**
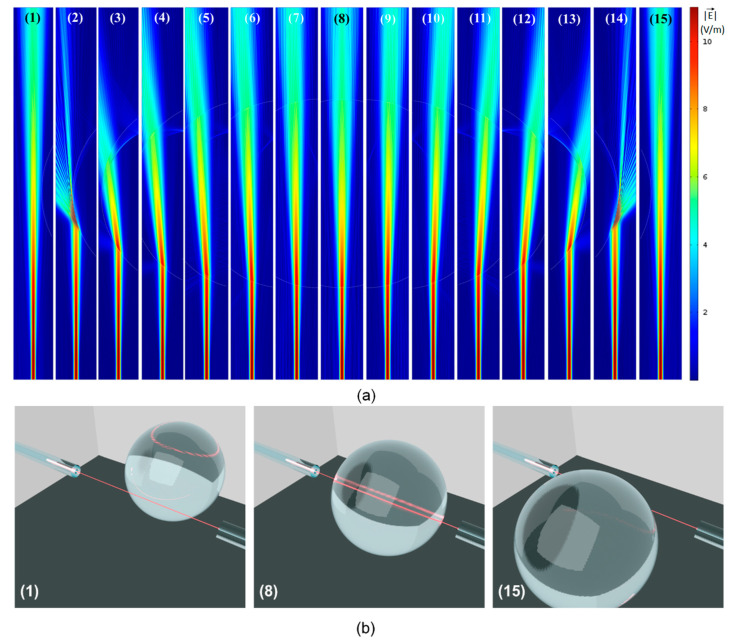
(**a**) Magnitude of the electric field vector for 15 different positions of the input/output fibers with respect to the droplet. The droplet moves 20 µm between positions. (**b**) Illustration of the positions 1, 8, and 15-of the input/output fibers with respect to the droplet.

**Figure 5 sensors-22-04999-f005:**
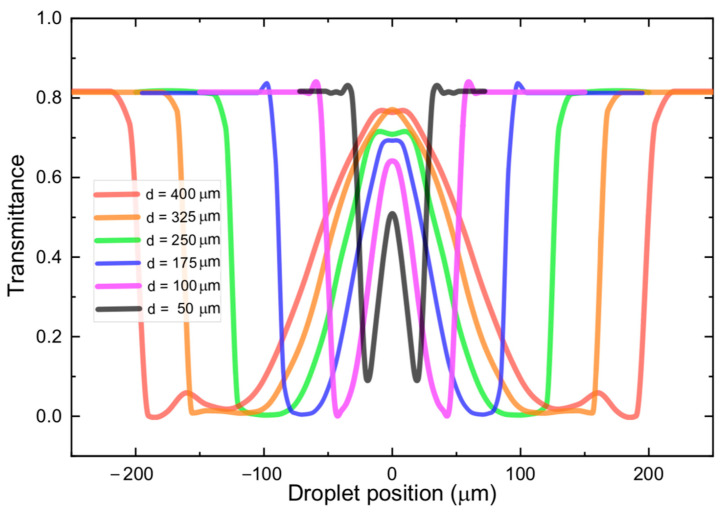
Modelled transmittance as a function of droplet position. Droplet position 0 µm, corresponds to the situation in which the beam of light is going straight to the middle of the droplet. The six different curves correspond to droplets with diameters ranging from 50 µm (black) up to 400 µm (red).

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
