# Peer review of "Real Time Water-In-Oil Emulsion Size Measurement in Optofluidic Channels"

_sensors, 2022, doi:10.3390/s22134999_

Round 1

Reviewer 1 Report

Review Manuscript ID: sensors-1774805     Type: article

Title: Real time Water-in-Oil emulsion size measurement for optofluidic applications

Authors: Juliana De Novais Schianti , Igor Yamamoto Abe , Marco Isaías Alayo Chávez , Daniel Orquiza de Carvalho

The authors present in this manuscript an optical platform for the detection and size measurement of emulsion droplets in real time in optofluidic platforms. This platform is not new and it is similar to the one used in ref. 21. The main contribution of the authors is the electromagnetic modeling used to understand the physical phenomena behind the optical transmission model obtained for the emulsion droplets. It is a good article worth publishing after considering some points listed above.

1. Introduction: 

The introduction is very well written and clarifies the importance of the emulsion droplet for smart drugs and DNA analysis to ensure low power consumption and a high level of control over physical and chemical properties. 

·      References 9 and 11 to 15 may be updated. The sentence in line 36: “Recently, microfluidic platforms in which the entire PCR process is realized entirely on chip have been proposed [9–15]. References are from 2007, 2008 and 2009.

·      In line 60: “input fiber (10 um in diameter)” but in line 101 “a core diameter of 8.2 um”

·      In line 40: The detection limit for DNA analysis (IS ?) much smaller than in bulk solutions and the size of the water-in-oil droplets used directly affects this detection limit.

·       

2. Device fabrication

·      Line 89: “a core diameter of 10 um”

3. Experimental setup and results

·      The authors use an SMF-28 fiber at 632.8 nm, this fiber is multimode at this wavelength. Why didn't you use an SM600 for example?

4. Electromagnetic modelling and discussion

·      In line 153: “the fundamental mode of the single mode input fiber was set up as excitation”. The SMF-28 fiber is multimode at 632.8 nm. Is it possible to simulate different modes of propagation as excitation?

Author Response

A version of the response in which the reviewer's comments and the author's responses are highlighted in different colors is attached in .docx format.

Comments and Suggestions for Authors: The authors present in this manuscript an optical platform for the detection and size measurement of emulsion droplets in real time in optofluidic platforms. This platform is not new and it is similar to the one used in ref. 21. The main contribution of the authors is the electromagnetic modeling used to understand the physical phenomena behind the optical transmission model obtained for the emulsion droplets. It is a good article worth publishing after considering some points listed above.

  1. Introduction:

- The introduction is very well written and clarifies the importance of the emulsion droplet for smart drugs and DNA analysis to ensure low power consumption and a high level of control over physical and chemical properties.

Response:

We thank the reviewer.

- References 9 and 11 to 15 may be updated. The sentence in line 36: “Recently, microfluidic platforms in which the entire PCR process is realized entirely on chip have been proposed [9–15]. References are from 2007, 2008 and 2009.

Response:

References 9 and 11-15 have been updated by more recent references. Details are presented below:

  1. Zhu, C.; Hu, A.; Cui, J.; Yang, K.; Zhu, X.; Liu, Y.; Deng, G.; Zhu, L. A Lab-on-a-Chip Device Integrated DNA Extraction and Solid Phase PCR Array for the Genotyping of High-Risk HPV in Clinical Samples. Micromachines 2019, 10, 537, doi:10.3390/mi10080537.
  2. Shen, K.-M.; Sabbavarapu, N.M.; Fu, C.-Y.; Jan, J.-T.; Wang, J.-R.; Hung, S.-C.; Lee, G.-B. An Integrated Microfluidic System for Rapid Detection and Multiple Subtyping of Influenza A Viruses by Using Glycan-Coated Magnetic Beads and RT-PCR. Lab Chip 2019, 19, 1277–1286, doi:10.1039/C8LC01369A.
  3. Yang, C.; Deng, Y.; Ren, H.; Wang, R.; Li, X. A Multi-Channel Polymerase Chain Reaction Lab-on-a-Chip and Its Application in Spaceflight Experiment for the Study of Gene Mutation. Acta Astronautica 2020, 166, 590–598, doi:10.1016/j.actaastro.2018.11.049.
  4. Yin, Z.; Ramshani, Z.; Waggoner, J.J.; Pinsky, B.A.; Senapati, S.; Chang, H.-C. A Non-Optical Multiplexed PCR Diagnostic Platform for Serotype-Specific Detection of Dengue Virus. Sensors and Actuators B: Chemical 2020, 310, 127854, doi:10.1016/j.snb.2020.127854.
  5. Kang, B.-H.; Lee, Y.; Yu, E.-S.; Na, H.; Kang, M.; Huh, H.J.; Jeong, K.-H. Ultrafast and Real-Time Nanoplasmonic On-Chip Polymerase Chain Reaction for Rapid and Quantitative Molecular Diagnostics. ACS Nano 2021, 15, 10194–10202, doi:10.1021/acsnano.1c02154.
  6. Kevadiya, B.D.; Machhi, J.; Herskovitz, J.; Oleynikov, M.D.; Blomberg, W.R.; Bajwa, N.; Soni, D.; Das, S.; Hasan, M.; Patel, M.; et al. Diagnostics for SARS-CoV-2 Infections. Nat. Mater. 2021, 20, 593–605, doi:10.1038/s41563-020-00906-z.

- In line 60: “input fiber (10 um in diameter)” but in line 101 “a core diameter of 8.2 um”

Response:

We modified the value of the line 60 for “8.2 um” as stated in the manufacturer's specs of the SMF-28 optical fiber

https://www.newport.com/p/F-SMF-28

- In line 40: The detection limit for DNA analysis (IS ?) much smaller than in bulk solutions and the size of the water-in-oil droplets used directly affects this detection limit.    

Response:

We thank the reviewer. We added the word “is” in the sentence.

  1. Device fabrication

- Line 89: “a core diameter of 10 um”

Response:

We modified the value of the line 89 for “8.2 um” such as explained previously.

  1. Experimental setup and results

- The authors use an SMF-28 fiber at 632.8 nm, this fiber is multimode at this wavelength. Why didn't you use an SM600 for example?

Response:

The reviewer’s comment is correct. This optical fiber presents 5 modes at 632.8 nm-wavelength (result obtained using FEM simulations). However, despite the input optical fiber have been multimode, it did not compromise the measurements carried out in this work. Our objective was to inject an incoming light beam with a diameter relatively smaller than the diameter of the measured droplet. As the diameter of the smallest measured droplet was 60 um and the diameter of the input fiber core was 8.2 um, we guaranteed thus the proposed objective. The negligible influence of the presence of other modes was also corroborated by the results obtained; the simulation results matched perfectly with the experimental results. Even so, to avoid confusion in the manuscript, we changed the word “single mode input optical fiber” by “8.2um core diameter input optical fiber”

  1. Electromagnetic modelling and discussion

 - In line 153: “the fundamental mode of the single mode input fiber was set up as excitation”. The SMF-28 fiber is multimode at 632.8 nm. Is it possible to simulate different modes of propagation as excitation?

Response:

Unfortunately, it was not possible to do simulations with all the modes at the same time, but it was possible to do simulations with the higher modes separately and the results remained almost unchanged. So, we decided to present only the results of the fundamental mode. In the manuscript the sentence: “simulations were also performed with higher modes for the input optical fiber, but the results were similar to those obtained for the fundamental mode” was added to better clarify about the simulation of other modes.

Finally, we would like to mention that the English was revised in all the manuscript and the changes were done using the “track changes” function of the MS Word.

Reviewer 2 Report

The authors present a sensor based in a 3-layer device. There is a T-channel in which the water-in-oil emulsion can be monitored. The input signal is connected to a single mode fiber transmitting a red-light laser, while the output signal is collected by a multi-mode fiber that is connected to a light detector, and processed by Labview. The authors demonstrated an interesting device; however, the authors must clarify or act in the provided comments.

Comment 1. In line 9, there is a capitalized letter in the word Electromagnetic, the same typo issue along the manuscript.

Comment 2. In line 46, please describe the RF acronym.

Comment 3. In line 50, use analyzed instead of “analysed”. Please review the grammar all over the manuscript.

Comment 4. It is possible to attach PC connectors to the device to avoid fiber optic ruptures or insertion losses?

Comment 5. In line 97, quoted ´cut´ word needs to be replaced by “cut”

Comment 6. In your experiment what kind of water and oil are used?

Comment 7. In line 89, the authors mention that the input fiber has a core diameter of 10 um; however, in line 101, the authors mention that were using an input fiber with a core diameter of 8.2 um, so there is a mismatch of around 8 % in the core diameter, a non-negligible rate.

Comment 8. From Figure 2, add a discussion about the fluctuation reflected in the transmitted power for the different samples even if the output is normalized. It is possible to present this sensor in transmitted power regime?

Comment 9. I am doubtful of the presentation of Figure 3. Why use arbitrary units if previously in Figure 2 presented measurements in a span time of around 2 sec? Obviously, the measured FWHM is related to time.

Comment 10. What is the software used to model the FEM?

Comment 11. The authors assume total reflection in droplets, but could be there any indicator of whispery gallery propagation?

Comment 12. What is the minimum detectable droplet size of the presented sensor and its sensitivity?

Author Response

A version of the response in which the reviewer's comments and the author's responses are highlighted in different colors and graphs obtained in simulations are shown is attached in .docx format.

Reviewer 2:

The authors present a sensor based in a 3-layer device. There is a T-channel in which the water-in-oil emulsion can be monitored. The input signal is connected to a single mode fiber transmitting a red-light laser, while the output signal is collected by a multi-mode fiber that is connected to a light detector, and processed by Labview. The authors demonstrated an interesting device; however, the authors must clarify or act in the provided comments.

We appreciate the reviewer’s comments on our manuscript and we will try to address each comment in the best possible way.

Comment 1. In line 9, there is a capitalized letter in the word Electromagnetic, the same typo issue along the manuscript.

Response:

We acknowledge the mistake and we have changed the word ‘Electromagnetic’ to ‘electromagnetic’ in the manuscript.

Comment 2. In line 46, please describe the RF acronym.

Response:

RF is an acronym for radiofrequency. We have made this explicit in the first instance of the term in the manuscript (Line 46, paragraph 4 of the introduction).

Comment 3. In line 50, use analyzed instead of “analysed”. Please review the grammar all over the manuscript.

Response:

We have corrected the mistake and reviewed the Manuscript thoroughly.

Comment 4. It is possible to attach PC connectors to the device to avoid fiber optic ruptures or insertion losses?

Response:

Yes, it would be possible to insert physical contact connectors by splicing them into the fiber. The input fiber is already connectorized and doing this to the output fiber would be perfectly feasible.

We have introduced the phrase “The input fiber is connected to the laser using a PC connector and it would be possible to splice fibers with different types of connectors, such as PC connectors, to both these fibers in order to use lasers and detectors of different types.”  in the last line of the last paragraph of section 2 – Devices fabrication.

Comment 5. In line 97, quoted ´cut´ word needs to be replaced by “cut”

Response:

We thank the reviewer. We have made this change to the text.

Comment 6. In your experiment what kind of water and oil are used?

Response:

This is a very good observation of information that is missing in the manuscript. We have changed the word ‘oil’ inside the parenthesis in the first paragraph of section 3 (Experimental setup and results) to ‘soybean oil’. We have also changed the word “water” inside the parenthesis in the first paragraph of section 3 (Experimental setup and results) to ‘de-ionized water’.

Comment 7. In line 89, the authors mention that the input fiber has a core diameter of 10 um; however, in line 101, the authors mention that were using an input fiber with a core diameter of 8.2 um, so there is a mismatch of around 8 % in the core diameter, a non-negligible rate.

Response:

This was in fact a careless mistake. The input fiber used was SMF-28, and the core diameter is 8.2um. We have corrected line 89 and replaced “10um” with “8.2um”, as well as other instances in the manuscript in which this mistake occurred.

Comment 8. From Figure 2, add a discussion about the fluctuation reflected in the transmitted power for the different samples even if the output is normalized. It is possible to present this sensor in transmitted power regime?

Response:

Addressing the question in the last part of the reviewer’s comment, we assume that the question is if it would be possible to present the sensor in reflected power regime, since Figure 2 presents the transmitted power (if we misunderstood the question, we apologize and kindly ask the reviewer to clarify). It would indeed be possible to have this sensor to work in the reflected power regime, since we used normal incidence, but this would require additional components in the setup, such as a circulator or at least a directional coupler connected at the input power to separate the reflected mode from the incident mode. We have added the suggested discussion to the text by adding an additional paragraph to section 3 (3rd paragraph of this section in the revised version of the manuscript):

“The reason why the graphs in Figure 2 present different amplitudes of the M-shaped patterns, even for a normalized transmitted power, is attributed to the fact that, especially for smaller droplets, it is impossible to control the droplet position along the out-of-plane (z-direction) with respect to the microchannel surface, in our device. For this reason, for some of the samples, the beam might go through a point which is much closer to the ‘equator’ of the droplet, and for others the transmitted beam might go through a point which is farther from the equator. In the latter case, due to refraction, the beam would be deflected vertically, away from the output fiber, creating a drop in the transmitted power. To account for this in electromagnetic modelling would require full 3D-simulations, which unfortunately we are not able to perform due to limitations in our computational resources. For this reason, as will be discussed in the next section, we have performed 2D simulations in order to understand these phenomena. It would also be possible to have this sensor to work in the reflected power regime, since we used normal incidence, but this would require additional components in the setup, such as a circulator or at least a directional coupler connected at the input power to separate the reflected mode from the reflected mode.”

Comment 9. I am doubtful of the presentation of Figure 3. Why use arbitrary units if previously in Figure 2 presented measurements in a span time of around 2 sec? Obviously, the measured FWHM is related to time.

Response:

The reviewer is correct, the units should not be arbitrary, but actually millimeters. The units of the abscissa of figure 2 are second but, stated in the text, the abscissa in figure 3 are de-normalized with respect to flow velocity, so the actual units are millimeters. This was corrected in the x-axis label of Figure 3 which now reads: “FWHM of transmittance curve (mm).

Comment 10. What is the software used to model the FEM?

Response:

The software Comsol Multiphysics was used for FEM analysis. We have provided this information in the manuscript by adding the phrase “using the commercial software Comsol Multiphysics” after referencing the FEM in the first paragraph of section 4 (Electromagnetic modelling and discussion).

Comment 11. The authors assume total reflection in droplets, but could be there any indicator of whispery gallery propagation?

Response:

This would be theoretically possible, but we have not detected it in our electromagnetic simulations. Perhaps the mode is attenuated at the used wavelength due to absorption, but no whispering gallery mode was seen. This could be an interesting follow up work for the present work.

Comment 12. What is the minimum detectable droplet size of the presented sensor and its sensitivity?

Response:

The minimum detectable size (droplet diameter) was 10um, this information was obtained via simulations, since our device is not able to generate droplets smaller than ~60um. In the following graph we show the transmittance obtained through simulations for droplets diameters of 5um and 10um. As it can be seen, A FWHM of ~10um can be detected for the curve corresponding to a droplet diameter of 10um. For the 5um-wide droplet, on the other hand, since it is smaller (half the approximate size) of the beam waist it does not deflect the beam on its entirety. For this reason, an oscillating pattern is present in the corresponding transmittance curve, which makes it difficult to discern the droplet diameter with any sort of precision.

The following sentence was added to the end of the last paragraph of section 4: “The minimum droplet that was detectable in our simulations was 10 µm, which corresponds to dimensions close to the beam diameter exiting the input fiber.”

Finally, we would like to mention that the English was revised in all the manuscript and the changes were done using the “track changes” function of the MS Word.

Reviewer 3 Report

This manuscript reports an optofluidic method to measure water droplet size. As mentioned by the authors, the main contribution of this work is to investigate the mechanism of the upside-down M-shaped “Transmission vs Time” curve. But the phenomena itself (upside-down M-shaped curve) is quite intuitive, and the investigation is also fairly simple. In addition, the advantage of the proposed system is not clear. So, I believe this manuscript is not suitable to publish in Sensors. Below are some detailed comments.

  1. The title: “xx measurement for optofluidic applications”. But I didn’t see any demonstration of optofluidic applications.
  2. I notice a GRIN multimode fiber was used as the receiver. Can you explain why “GRIN multimode fiber”, rather than a common multimode fiber was used?
  3. In line 60-61, the authors claimed their system is better than the one in Ref [21] because they are using fibers with smaller cores. I would suggest the authors extend the experiment by comparing your system performance with the one in Ref [21].
  4. In line 81, the authors mentioned the channel width is 250 um. Based on my personal experience, it is not easy to fabricate such narrow channels with CO2 laser cutter. Are there any microscope images of these channels? What is the surface roughness of these channels? Will the surface roughness induce significant light scattering?
  5. In the fabrication part, the authors did not mention how the three acrylic layers were bonded together. 
  6. In Fig4(a), there should be a color bar.
  7. In the model to analyze the light transmission, the droplet was assumed to locate at the center of microchannel, right? How would their position variation affect the results? Here the position variation refers to both in-plane (x-y axis) and out-of-plane (z-axis) variations. The authors need to extend this discussion extensively.. 

Author Response

A version of the response in which the reviewer's comments and the author's responses are highlighted in different colors and graphs  are shown is attached in .docx format.

Reviewer 3:

Comments and Suggestions for Authors: This manuscript reports an optofluidic method to measure water droplet size. As mentioned by the authors, the main contribution of this work is to investigate the mechanism of the upside-down M-shaped “Transmission vs Time” curve. But the phenomena itself (upside-down M-shaped curve) is quite intuitive, and the investigation is also fairly simple. In addition, the advantage of the proposed system is not clear. So, I believe this manuscript is not suitable to publish in Sensors. Below are some detailed comments.

  1. The title: “xx measurement for optofluidic applications”. But I didn’t see any demonstration of optofluidic applications.

Response:

The reviewer is absolutely correct, we did not actually demonstrate any specific application. We have therefore changed the title to read: “Real time Water-in-Oil Emulsion Size Measurements in Optofluidic Channels”.

  1. I notice a GRIN multimode fiber was used as the receiver. Can you explain why “GRIN multimode fiber”, rather than a common multimode fiber was used?

Response:

We appreciate the observation. We believe that no real gain or loss would have been obtained by using a multimode fiber instead of the GRIN fiber. The reason we used a GRIN fiber is simply because it is what we had available in our laboratory. The point of using a fiber with a larger core as an output fiber is related to the fact that, as can be seen in the simulations (Figure 4a), the beam which has an approximate diameter of 10um right at the exit point of the input fiber, reaches the output fiber with a diameter of ~50um. Therefore, a multimode fiber (be it GRIN or regular) would be able to capture this light more effectively than a single mode fiber.

  1. In line 60-61, the authors claimed their system is better than the one in Ref [21] because they are using fibers with smaller cores. I would suggest the authors extend the experiment by comparing your system performance with the one in Ref [21].

Response:

We would like very much to present this comparison, but unfortunately, ref [21] does not present the droplet dimensions, but rather present the signal as a function of time without even specifying the corresponding flow rates. This makes it very difficult to even estimate the size of the droplet in [21]. The reason we state that, potentially, our system would be able to measure smaller droplets is because our beam diameter is somewhere in between 10um (close to the input fiber) and 50 um (close to the output fiber). Since in [21] the fiber diameter (both input and output) is 100um, which is much larger, we supposed that the minimum droplet diameter that can be measures is larger, seen as there should be a relationship between beam diameter and the minimum value of droplet diameter it can resolve. Given that we do not have droplet diameter data, we agree that this is not a precise comparison. For this reason we have changed the part of the last paragraph that deals with this subject to read: “Because the core diameter of both the input fiber (8.2 µm in diameter) and output fiber (50 µm in diameter) are significantly smaller than the ones used in previous works (~100 µm in diameter) [20], we presume that the investigated device could be used to measure smaller droplets, although this cannot be backed by experimental data since the droplet diameter is not specified in that work.”

  1. In line 81, the authors mentioned the channel width is 250um. Based on my personal experience, it is not easy to fabricate such narrow channels with CO2 laser cutter. Are there any microscope images of these channels? What is the surface roughness of these channels? Will the surface roughness induce significant light scattering?

Response:

The figure below shows an optical microscope image of the 250um-wide channel. The width measured using image analysis software is 245um, so it seems that the laser is able to do a reasonable job. The dark line on both walls of the channel is because of the slanted sidewalls (the microscope seems to be a little bit slanted). The roughness was estimated to be around 13um. This is not a big problem for our system since both input and output fibers are positioned in such a way that light impinges normally upon these surfaces, and the fraction of power which is actually scattered does not seem to be too large, since the power at the output fiber is of the order of hundreds of microWatts and the laser diode power is 1 mW. We have introduced the following sentence in the  second paragraph of section 2:

“The roughness associated with the channel sidewalls was estimated to be around 13 µm using image analysis. This is not a big problem for our system since both input and output fibers are positioned in such a way that light impinges normally upon these surfaces, and the fraction of power which is actually scattered does not seem to be large, since the power at the output fiber is of the order of hundreds of µWatts and the laser diode power is 1 mW.”

  1. In the fabrication part, the authors did not mention how the three acrylic layers were bonded together.

Response:

This is a very good and important point, the acrylic layers were bonded using UV cured adhesive. The following sentence was added to the second paragraph of section 2 (device fabrication):

“The three layers shown in Figure 3 were bonded using UV curing adhesive (Loxeal, 30-21), which was exposed to light from a 36W UV lamp for several minutes.”

  1. In Fig4(a), there should be a color bar.

Response:

We have added a color bar to Figure 4(a), as suggested by the reviewer.

  1. In the model to analyze the light transmission, the droplet was assumed to locate at the center of microchannel, right? How would their position variation affect the results? Here the position variation refers to both in-plane (x-y axis) and out-of-plane (z-axis) variations. The authors need to extend this discussion extensively..

Response:

We agree and we have added the following paragraph addressing this issue (3rd paragraph of section 3 in the new version of the manuscript:

“The reason why the graphs in Figure 2 present different amplitudes of the M-shaped patterns, even for a normalized transmitted power, is attributed to the fact that, especially for smaller droplets, it is impossible to control the droplet position along the out-of-plane (z-direction) with respect to the microchannel surface, in our device. For this reason, for some of the samples, the beam might go through a point which is much closer to the ‘equator’ of the droplet, and for others the transmitted beam might go through a point which is farther from the equator. In the latter case, due to refraction, the beam would be deflected vertically, away from the output fiber, creating a drop in the transmitted power. To account for this in electromagnetic modelling would require full 3D-simulations, which unfortunately we are not able to perform due to limitations in our computational resources. For this reason, as will be discussed in the next section, we have performed 2D simulations in order to understand these phenomena. It would also be possible to have this sensor to work in the reflected power regime, since we used normal incidence, but this would require additional components in the setup, such as a circulator or at least a directional coupler connected at the input power to separate the reflected mode from the reflected mode.”

We have addressed this issue by inserting scale bars in all of the optical microscope pictures of the droplets.

Finally, we would like to mention that the English was revised in all the manuscript and the changes were done using the “track changes” function of the MS Word.

Reviewer 4 Report

A platform for real time emulsion droplet detection and size measurement in optofluidic platforms has been demonstrated in this research, and the paper has been well organized. But the following concerns need to be improved before its online publication.

1.      A laser diode emitting 1 mW of optical power at 632.8 nm, coupled to an optical fiber with a core diameter of 8.2 µm (SMF-28) was used to couple light to one side of the micro-channel (input fiber in Figure 1(a)). However, the transmitted power in dependence of flow rates fluctuates a lot between 900 and 700 as shown in Figure 2, what causes the transmitted power fluctuation?

2.      How about the size of captured fluid droplets as shown in Figure 3? Please insert scale bars in optical microscopy images.

Author Response

A version of the response in which the reviewer's comments and the author's responses are highlighted in different colors is attached in .docx format.

Reviewer 4:

Comments and Suggestions for Authors: A platform for real time emulsion droplet detection and size measurement in optofluidic platforms has been demonstrated in this research, and the paper has been well organized. But the following concerns need to be improved before its online publication.

  1. A laser diode emitting 1 mW of optical power at 632.8 nm, coupled to an optical fiber with a core diameter of 8.2 µm (SMF-28) was used to couple light to one side of the micro-channel (input fiber in Figure 1(a)). However, the transmitted power in dependence of flow rates fluctuates a lot between 900 and 700 as shown in Figure 2, what causes the transmitted power fluctuation?

Response:

The reason for the power fluctuation when the droplet is not in the optical path has to do with the large dependence of the power emitted by the laser diode with respect to temperature. This is why, for the six different measurements shown in Figure 2, the horizontal line fluctuates between 700 and 900 a.u. between one measurement and the other. We have added the following sentence to the end of the first paragraph of section 3:

“The reason for the power fluctuation observed when the droplet is not in the optical path (baseline fluctuation) has to do with the large dependence of the power emitted by the laser diode with respect to temperature. This is why, for the six different measurements shown in Figure 2, the horizontal line fluctuates between 700 and 900 a.u. between one measurement and the other. This could be addressed by temperature control of the emitted power.”. Another important point to mention is that only the FWHM is used to measure the droplet diameter, then any variation in the amplitude should not significantly affect the FWHM value.

  1. How about the size of captured fluid droplets as shown in Figure 3? Please insert scale bars in optical microscopy images.

Response:

We have addressed this issue by inserting scale bars in all of the optical microscope pictures of the droplets.

Finally, we would like to mention that the English was revised in all the manuscript and the changes were done using the “track changes” function of the MS Word.

Round 2

Reviewer 2 Report

I am satisfied with the responses

Reviewer 3 Report

The authors have clarified my concerns. So I would recommend its publication in Sensors

Reviewer 4 Report

No comments at this moment.